# Ergogenic and Physiological Effects of Sports Supplements: Implications for Advertising and Consumer Information

**DOI:** 10.3390/nu17162706

**Published:** 2025-08-21

**Authors:** Pedro Estevan Navarro, Cristina González-Díaz, Rubén García Pérez, Angel Gil-Izquierdo, Carlos Javier García, Daniel Giménez-Monzo, Alejandro Perales, José Miguel Martínez Sanz

**Affiliations:** 1Faculty of Health Sciences, University of Alicante, 03690 Alicante, Spain; pedroestevandn@gmail.com (P.E.N.);; 2Research Group on Applied Dietetics, Nutrition and Body Composition (DANuC), University of Alicante, 03692 Alicante, Spain; josemiguel.ms@ua.es; 3Psychology and Social Communication Department, Faculty of Economics and Business, University of Alicante, 03690 Alicante, Spain; 4Research Group on Quality, Safety, and Bioactivity of Plant Foods Group, Department of Food Science and Technology, CEBAS-CSIC, University of Murcia, 30100 Murcia, Spain; cjgarcia@cebas.csic.es; 5Department of Community Nursing, Preventive Medicine and Public Health and History of Science Health, University of Alicante, 03690 Alicante, Spain; dgimenez@ua.es; 6Research Group on Food and Nutrition (ALINUT), University of Alicante, 03692 Alicante, Spain; 7Communication Sciences and Sociology, Faculty of Communication Sciences, Rey Juan Carlos University, 28933 Madrid, Spain; aperales@auc.es; 8Nursing Department, Faculty of Health Sciences, University of Alicante, 03690 Alicante, Spain

**Keywords:** sports supplements, claims, ergogenic effects, physiological effects, advertising, consumer information, regulatory framework, athletic performance

## Abstract

Background: The use of sports supplements has increased significantly in athletic contexts, raising the need to evaluate their efficacy, safety, regulatory status, and communication practices. Objective: This study aimed to describe and synthesize the ergogenic and physiological effects of Australian Institute of Sport (AIS) Category A performance supplements. Methods: A descriptive and observational study was conducted, collecting and analyzing information from systematic reviews and position statements related to performance supplements, including caffeine, creatine, β-alanine, nitrate/beetroot juice, sodium bicarbonate, and glycerol. Results: Caffeine and creatine are the only supplements with authorized health claims. However, β-alanine, nitrates, sodium bicarbonate, and glycerol show positive ergogenic effects supported by strong evidence, especially in endurance, strength, high-intensity, and aquatic sports. However, these substances lack regulatory approval, and only a small proportion of commercial products comply with current legislation. Conclusions: While performance supplements may enhance athletic performance when used alongside proper nutrition and scientific guidance, their effectiveness is not always consistent or assured. This review highlights the urgent need to update regulatory frameworks, harmonize labeling standards, and promote ethical marketing to safeguard consumers and support sports and nutrition professionals.

## 1. Introduction

Currently, there is a wide variety of foods available for consumption, each with specific characteristics and specific functions in the body, among which we find food supplements. At the regulatory level, they do not have a specific legal category, being regulated by Directive 2002/46/EU (its national transposition, as consolidated legislation, is carried out through Royal Decree 1487/2009 of 26 September 2009, on food supplements) which defines them in Article 2 as “foodstuffs intended to supplement the normal diet and consisting of concentrated sources of nutrients or other substances that have a nutritional or physiological effect” [1]. Within the context of sports nutrition, a frequently used option to supplement the daily diet is food supplements, also called sports supplements. Specifically, sports supplements (hereafter SSs) are designed for individuals aiming to enhance physical performance, optimize recovery after exercise, or promote their general state of health [2]. The International Olympic Committee (IOC) defines SSs as “a food, food component, nutrient, or non-food compound that is intentionally ingested in addition to the customarily consumed diet for the purpose of achieving a health or performance benefit” [3]. Despite the strict regulatory framework for nutrition and health claims in the European Union—established through Regulation (EC) No 1924/2006 and guided by EFSA’s scientific opinions—many food supplement companies continue to use unapproved or borderline claims while awaiting official authorization. This creates a communication gap between scientific evidence and consumer-facing marketing language. The present study addresses this gap by evaluating how performance-enhancing supplement claims are communicated across digital platforms, and to what extent these claims align with EFSA’s current assessment process and regulatory status [3,4].

SS products are marketed in multiple formats, such as capsules, tablets, powders, or beverages [2]. Their accessibility and ease of consumption explain their high prevalence, and it is estimated that between 40% and 100% of athletes use some type of supplement in their regular diet. Among the most consumed are caffeine, creatine, nitrates (or beetroot juice), sodium bicarbonate, and β-alanine [4,5,6,7]. Moreover, these SSs have considerable scientific support for their efficacy and safety [3,8,9,10,11].

SS consumption is heavily influenced by industry marketing and advertising efforts aimed at maximizing consumer reach [4]. This trend has been reinforced by the development of new technologies and the increasing ease of purchasing products through the Internet [12]. It is estimated that the value of the sports supplement market in Europe reached EUR 12 billion between 2018 and 2019, and it is projected that it could reach EUR 45 billion by 2026 [13].

In order to be marketed, SSs must comply with specific regulations governing aspects such as labeling, advertising, and the truthfulness of the information provided to the consumer [14]. At the European level, current legislation is framed in Regulation (EU) No. 1169/2011 of the European Parliament on food information provided to the consumer. It should be noted that special mention is made of the form and content of the presentation of the commercial communication of these types of components, which must be easily understood by the so-called “average consumer”, who is defined as one who “is reasonably well informed and reasonably observant and circumspect, taking into account social, cultural and linguistic factors”. This definition is established in REGULATION (EC) No. 1924/2006 on nutrition and health claims made on foods and referred to in Regulation (EU) No. 1169/2011 through consideration 41.

Current legislation requires commercial communication regarding SSs to be clear, truthful, not misleading, and easily understood by the average consumer. However, there are already studies that highlight the lack of transparency in the composition of supplements, which entails risks to health and consumer confidence [15]. On the other hand, within the communication campaigns used by the brands of these products, whose consumption is on the rise, is the use of health claims within the framework of the “halo effect”; that is, creating a good brand impression through the use of “Fitness Influencers” [16]. In fact, an increasing number of companies are using these “referents” in their communication strategies, given the connotations of values such as physical care and healthy lifestyles that permeate the brands [16].

One of the key elements in the advertising communication of SSs is the use of Health Claims, defined by Regulation 1924/2006 as “any statement on labels, advertising or other marketing products that claims that the consumption of a particular food may benefit health” [17]. The European Food Safety Authority (hereinafter EFSA), the authority that evaluates the request for claims by issuing scientific opinions, prepares a Community register of Health Claims, which lists those authorized claims together with their conditions of use [18].

Similarly, in the area of SS, the EFSA also receives applications related to sports performance, rejecting those for which the scientific evidence has not been conclusive.

Currently, health claims approved by the European Commission related to sports supplements refer mainly to nutrients such as protein, carbohydrates, electrolytes, caffeine, and creatine [18]. However, some supplements that have strong scientific evidence for use in specific sport contexts and under evidence-based protocols [8,19], such as sodium bicarbonate, β-alanine, nitrates, and glycerol, do not have a Commission-approved claim, despite having a favorable opinion from the EFSA.

Therefore, the aim of this study is to describe and compile a list of the ergogenic and physiological effects of the most evidenced sports supplements in relation to position papers and systematic reviews.

The hypothesis is that most agencies and recent reviews with or without meta-analyses will agree on the majority of doses, protocols, mechanisms of action, etc., of the SSs studied.

With all of the information gathered, a series of guidelines will be proposed, by way of suggestions and guidelines, of the implications and potential applications that this work may have for the elaboration of messages in the context of advertising communication and consumer protection.

## 2. Materials and Methods

### 2.1. Type of Study

This study was a narrative review with practical applications based on the analysis of the protocol of use and effects of the consumption of SSs framed as category A performance supplements, according to the Australian Institute of Sport (AIS) [8]. The effects of these supplements were analyzed based on consensus documents and recent systematic reviews with meta-analyses.

### 2.2. Sample Selection Strategy

Sample selection was based on the Australian Institute of Sport category A SSs [8]. This category includes caffeine, creatine, sodium bicarbonate, β-alanine, nitrates/beetroot juice, and glycerol, which are recommended for use with an individualized and event-specific protocol.

The Australian Institute of Sport’s (AIS) “emerging supplements”, or category B, were not included because, although they may have positive effects, we consider that they do not have sufficient evidence behind them to support their use, at least at present, at the same level as category A SSs. The decision to focus exclusively on category A supplements was based on their strong scientific support for use in specific sporting situations, with evidence-based protocols and endorsement from expert consensus. These supplements are appropriate for use by athletes following best-practice recommendations, unlike category B substances, which are only advised for use in research settings or under individual monitoring protocols. Including such emerging supplements would introduce greater uncertainty regarding protocols, effects, and regulatory implications, thus falling outside the aims of this review. The description of the effects on performance was carried out using scientific documents created by institutions such as the European Food Safety Authority (EFSA) [20], the International Society of Sports Nutrition (ISSN) [9], the International Olympic Committee (IOC) [3], the Academy of Nutrition and Dietetics (AND) [10], the AIS [8], and the Spanish Society of Sports Medicine (SEMD) [11]. In addition, recent systematic reviews with meta-analyses were included through the PubMed database, using the name of the supplement (e.g., “caffeine”) as a keyword and applying the systematic review and meta-analysis filter.

### 2.3. Criteria for Inclusion of Documents

Position/consensus papers and systematic reviews with meta-analyses in English and Spanish that grouped the dose, protocol of use, and the sport in which a given SS would be administered were included. The timeframe for position/consensus papers included studies from 2015 to offer the most recent and updated information according to scientific advances in the sports nutrition field, because the nutritional recommendations for athletes have undergone significant changes in the past 10 years [10]. Systematic reviews with meta-analyses published within the last 2 years were prioritized. In cases where no recent systematic reviews with meta-analyses were available for a given supplement and sport context, the most recent publications were considered, provided they met the inclusion criteria of relevance, methodological quality, and a focus on performance outcomes.

### 2.4. Data Extraction

The variables taken into account for the analysis of each SS were:Study or institution: entity or author(s) analyzing the SS according to the inclusion criteria.Sport: physical activity or exercise, subject to certain rules, where skills, dexterity, or physical strength are tested.Protocol of use: factors such as timing or conditions that determine the intake of an SS to avoid complications and enhance the effects.Dose: amount of SS to be ingested in one dose.Effect: physiological and/or ergogenic response on sports performance caused by the administration of an SS in a general or sport-specific manner.

Data extraction was conducted independently by two researchers (P.E.N. and R.G.P.) using a predefined template for all included documents. Disagreements were resolved through discussion and consensus, with the involvement of a third author (J.M.M.-S.) when necessary. This strategy aimed to reduce selection and interpretation bias. Extraction focused only on studies that met the inclusion criteria detailed above and emphasized practical and ergogenic outcomes.

### 2.5. Meeting of Experts and Development of Methodological Criteria

With the aim of unifying criteria in relation to the variables described in Section 2.4, a face-to-face meeting of the research team, composed of members of the research groups in Food and Nutrition (ALINUT) and in Applied Dietetics, Nutrition and Body Composition (DANuC) of the University of Alicante, was held. At that meeting, the ad hoc unification criteria were established by the team with respect to (1) dosage, general effects, and protocols of use, in which those characteristics common to or present in more than 75% of the documents included were selected; (2) categorization of the type of sport based on the classifications proposed by Louise Burke in 2010 [21]; and (3) marketing format, referring to “the physical form in which the product is formulated, dosed and presented to the consumer, which may influence its absorption, stability and acceptance” [3].

## 3. Results

We included seven systematic reviews with meta-analyses and five position statements, selected using predefined criteria. The information compiled on caffeine, creatine, sodium bicarbonate, β-alanine, nitrates/beetroot juice, and glycerol, in relation to the sports in which their use has been investigated, administration protocols, recommended doses, and general and specific effects on sports performance, is presented in Table A1, Table A2, Table A3, Table A4, Table A5 and Table A6. In general terms, the ergogenic effects of these supplements are mainly associated with endurance disciplines, intermittent high-intensity sports, and aquatic activities. The preparation of Table A1, Table A2, Table A3, Table A4, Table A5 and Table A6 was based on the most up-to-date consensus statements and the latest systematic reviews with meta-analyses and EFSA information available for each supplement [3,9,10,11,22,23,24,25,26,27,28,29,30,31,32,33,34,35,36,37]. Table A1, Table A2, Table A3, Table A4, Table A5 and Table A6 are included in Appendix A.

All of the supplements analyzed specify the dosage necessary to achieve a relevant ergogenic or physiological effect. With regard to health claims, only creatine and caffeine have EFSA-approved authorizations, while the other supplements present only claims supported by scientific evidence, but not officially recognized by the EFSA.

Table 1 presents the unification of criteria carried out by the research team in accordance with the methodology described above, and summarizes the data extracted from Table A1, Table A2, Table A3, Table A4, Table A5 and Table A6 for each supplement.

## 4. Discussion

The results obtained show that the sports supplements (SSs) analyzed have positive effects on performance in various disciplines, especially in endurance, strength, power, and intermittent sports. Although there are slight differences between the administration protocols proposed by different organizations and research, a common trend is observed in terms of doses and structures of use, which allows the establishment of recommendations applicable in practice. Caffeine and creatine are the only supplements that have health claims approved by the EFSA, while the rest, despite not having this regulatory backing, have demonstrated ergogenic effects based on scientific evidence. A wide variety of commercial formats (capsules, powders, gels, chewing gum, drinks) has also been found to be available, which facilitates their adaptation to the preferences and individual needs of athletes.

Caffeine supplementation has shown benefits in performance during aerobic, anaerobic, and team sport exercise [3,9,11,23,24], including a reduction in perceived exertion in both men and women [10,24]. The doses usually recommended range from 3 to 6 mg/kg administered before exercise, although its use during activity in lower doses is also contemplated [3,9,10,11,22,23,24]. Its availability in multiple formats (capsules, gels, chewing gum, and shots, among others) contributes to its practical applicability [10]. The EFSA has approved several health claims for caffeine, which gives greater regulatory certainty to its commercialization in Europe [29]. However, recent studies reveal that only 2.78% of the evaluated products fully comply with these criteria [4]. The effects of caffeine are explained through mechanisms such as adenosine receptor antagonism, increased endorphins, improved neuromuscular function, and vigilance [3]. These benefits have been observed in disciplines such as rowing, endurance sports, team sports, and combat sports [9,11,23,24,38]. Some authors consider that caffeine supplementation does not affect all people equally; for example, we know that caffeine metabolism differs significantly between individuals depending on genetics (e.g., CYP1A2 polymorphism and the moment of taking the caffeine (morning or afternoon)), which can affect its ergogenic effect or even lead to adverse reactions [39,40].

Creatine acts by increasing muscle phosphocreatine stores, which facilitates ATP resynthesis during high-intensity, short-duration exercise [3,9,10,11,22]. It has been linked to improvements in strength, power, muscle mass, and recovery capacity [3,9,10,11,22], as well as showing positive effects in women [24] and possible cognitive benefits [41]. Its use is well established through loading and maintenance protocols, with creatine monohydrate being the most recommended format [3,9,10,11,22].

β-alanine, a precursor of muscle carnosine, has demonstrated ergogenic efficacy in short-duration, high-intensity exercise, particularly in intermittent efforts [3,9,10,11,30,31]. Its usefulness has also been observed in team and racquet sports. Although protocols vary, there is consensus on the use of doses of 4 to 6 g/day for periods of between 4 and 24 weeks [3,9,10,11,30,31]. One of its most frequent adverse effects is paresthesia, especially from doses higher than 800 mg [40]. In addition to the effect on performance, it could reduce neuromuscular fatigue in the older adult population [9,11].

Female athletes may respond differently to creatine or β-alanine supplementation depending on hormonal status, yet most studies include mostly male participants [42].

Nitrates, or beetroot juice, act as precursors of nitric oxide (NO), whose synthesis occurs through the entero-salivary pathway: nitrates (NO_3_^−^), after ingestion, are reduced to nitrites (NO_2_^−^) by the action of oral bacteria, and subsequently to NO under hypoxic or acidosis conditions characteristic of physical exercise [3,9,10,11,32,33]. Nitric oxide participates in the modulation of skeletal muscle function, enhancing vasodilation, blood flow, mitochondrial efficiency, and muscle contraction. An improvement in type II fiber function, a decrease in ATP cost per contraction, and an increased tolerance to high-intensity exercise have also been described [3,10,11,32]. In addition, a reduction in oxygen demand at submaximal loads has been evidenced, implying an improvement in exercise economy [9]. These benefits have been observed mainly in endurance sports such as cycling, middle- and long-distance running, swimming, and in team sports such as soccer, rugby, and handball [9,11,13,32,33].

Sodium bicarbonate, thanks to its buffering effect, stabilizes intramuscular pH and promotes the release of H^+^, which delays fatigue during high-intensity, short-duration exercise [3,9,10,11,24,34,35]. It has demonstrated efficacy in team sports, combat, athletics, swimming, and cycling. Its most common presentation is in powder form, although formats such as tablets and gels have been developed. Protocols for use vary between acute doses (30–180 min before exercise) and fractionated protocols over several days, considering its possible adverse effect on the gastrointestinal tract [3,9,10,11,34].

Glycerol has shown mixed results, although a possible ergogenic effect is attributed to it through a state of hyperhydration that increases total body water volume and favors fluid retention [9,11,36,37]. Its use is more common in endurance sports, especially in hot environments, and it is marketed in powder and capsule formats.

Some of these supplements have shown positive effects beyond those of the AIS, such as creatine and beetroot juice, which have been shown to play a role in recovery, with creatine being more useful in power-focused efforts and beetroot juice in endurance-focused protocols [43].

Overall, the supplements analyzed, all classified in category A of the AIS, have well-documented ergogenic effects in different sports modalities. Although there are variations in the protocols of use, common patterns have been identified that allow the generation of general recommendations. This study can serve as a practical reference tool for athletes, coaches, and nutrition professionals by bringing together the current evidence on effects, doses, protocols, and possible side effects in an integrated manner [3].

The EFSA has approved health claims for caffeine and creatine based on safety criteria and scientific evidence [4,17,29]. In the case of caffeine, effects on endurance exercise performance and reduced perception of exertion are recognized, while for creatine, multiple claims related to strength, muscle mass, recovery, performance, and health have been authorized [28]. However, the limited agreement between marketed products and these criteria highlights the need for further regulatory control and possible updating of authorized claims.

The remaining supplements (sodium bicarbonate, nitrates, glycerol, and β-alanine) do not currently have EFSA-approved health claims, despite the scientific support given by bodies and experts in sports nutrition [3,9,10,11,22,44]. The claims attributed to these supplements are, in many cases, comparable to those recognized for caffeine and creatine, which justifies their reconsideration in the European regulatory framework. In this regard, it is noted that despite having multiple EFSA-approved health claims [28], only 25% of the supplements analyzed on the market partially or fully comply with these criteria. It should be noted that many of these claims were evaluated more than a decade ago, which may not fully reflect the current evidence.

Finally, the relevance of the commercial and advertising use of the ergogenic and physiological effects attributed to these supplements is highlighted [14], since the information declared on the labeling, web page, or technical data sheet significantly influences consumers’ purchasing decisions [14,45]. Despite the current regulations, there are legal gaps that make complete regulation difficult, since specific criteria have not been established for all supplements. Even in those products where clear regulations do exist, non-compliance has been detected, with differences between actual and declared ingredients, which can affect both performance and consumer health [3,46].

These results also show the need to update the regulation on communication regarding SSs from an advertising point of view given the potential lack of consumer protection, not only because of the information provided, but also because of the type of information provided in order to avoid confusion or lack of transparency with respect to the benefits of consumption. It should also be noted that, in this sense, progress is being made, since previous research [14] showed the gap between advertising communication and compliance with regulations. This study points in the opposite direction: although there are SSs that do have the support of the EFSA, the rest do not, despite having robust evidence that justifies the benefits that they propose.

### 4.1. Limitations of the Study

This study has several limitations that should be acknowledged. Although this work followed a structured and predefined methodology for the inclusion and synthesis of evidence, it was not a formal systematic review and did not follow PRISMA guidelines or include protocol registration. Although data extraction was conducted independently by two reviewers to minimize potential selection and interpretation bias, the synthesis of the results inherently involved some degree of subjectivity, especially when multiple documents provided divergent recommendations. A third reviewer resolved discrepancies when necessary, but this process cannot entirely eliminate bias in interpretation. Furthermore, the use of secondary sources (systematic reviews and position statements) may carry the biases of the original authors, such as publication bias, overrepresentation of certain populations (e.g., male athletes), or heterogeneity in study quality. The first limitation is that the analysis was based primarily on position papers and systematic reviews, which, although rigorous, may not capture the most recent individual trials or emerging evidence, for example with health claims approved by the EFSA. Second, the selection of supplements was limited to those classified as category A by the AIS, potentially excluding other SSs with growing scientific support. Third, the synthesis of protocols and dosages reflects general consensus, but does not consider inter-individual variability in response, which could influence efficacy and safety in practice. Fourth, there is a possible publication bias in the meta-analyses and the exclusion of non-English studies. Fifth, the regulatory analysis focused on the European context, limiting the generalizability of the findings to other regions with different legislative frameworks, like in the United States. Systematic reviews with more recent meta-analyses are needed to group the most current results of each SS in order to be able to modify, if necessary, part of the contents discussed in this study.

The exclusion of emerging supplements from AIS category B is another important limitation of the study. Although some of these products may have potential ergogenic effects, they currently lack sufficient scientific support and regulatory consensus for their general recommendation in sport settings. Their use is typically reserved for research contexts or under case-managed supervision, and thus falls outside the applied and practical focus of this manuscript. Future work may explore the evolving evidence base for these supplements as they transition toward broader acceptance.

It is important mention the potential commercial bias in studies or marketing materials analyzed, and the impact of underreporting adverse effects that can produce considerable variation in problems experienced by athletes.

Finally, an important limitation is that many systematic reviews include studies with small sample sizes or high variability in protocols. Moreover, some effects (e.g., of β-alanine or glycerol) are modest and context-dependent, yet often presented as generalizable.

### 4.2. Practical Applications

Numerous European and national organizations, including the EFSA, the Spanish Food Safety Agency (AESAN), and the Spanish Association of Food Supplements (AFEPADI), focus on consumer protection by promoting policies that seek regulation in order to provide security both in advertising claims and in the benefits that can be obtained after consumption. With this as a starting point, a series of guidelines are proposed as a guide for advertising communication and consumer information on SSs:-A harmonized standard should be proposed, at least at the European level, on the use of health claims and health properties and their use in advertising and labeling.-The minimum amount of information, at the level of benefits obtained, that should be specified for commercial communication should be determined.-Commercial communication on these types of products, in addition to complying with generic regulations, must take care of key aspects in both form and content to facilitate consumer information and understanding.-At the information level, specific labeling for SSs should be considered.-With regard to the claims for use, a list should be implemented at the regulatory level that “links” the claim and the minimum component that should make up the SS. In this regard, the parameters of REGULATION (EU) No. 432/2012 establishing a list of authorized health claims made on foods should be followed.-Given the online context and the boom in the use of fitness influencers for SS communication, a code of good practice between brands and these types of influencers should be promoted. It should be taken into account that their communication impacts not only their food intake, but also a lifestyle, have repercussions on the health of the people [47].-Media and advertising literacy for this type of product is key given the increase in its consumption and its projection in both traditional and online media.-The evidence presented in this study makes clear the effects, protocols, and sports related to the use of these well-studied supplements. This could serve as a tool of great interest for professionals and athletes because some supplements, such as beta alanine or glycerol, have effects that are dependent on a specific context such as climate or type of event, and some benefits/effects are still modest.

## 5. Conclusions

The papers reviewed show a high degree of consensus on the doses and mechanisms of action of AIS category A sports supplements (SSs), although there are some variations in administration protocols and observed effects, mainly attributable to differences between populations, methodologies, and sports disciplines. These variations are especially notable in supplements such as β-alanine and sodium bicarbonate, where supplementation times differ widely.

This work contributes to the systematization of this information, facilitating its practical application in the context of training and competition. It can also serve as a useful resource for dietitians, nutritionists, and coaches who require a general guide based on the supplements with the highest level of scientific support, promoting safer and more effective usage. Moreover, companies can use these findings as a guide for the development of appropriate statements with scientific support, along with information on protocol, dosage, etc.

At the regulatory level, regarding commercial communication and consumer protection, there is a legal vacuum that can lead to confusing communication that is not in line with the evidence, both in the way benefit claims are presented and in terms of consumption advice.

## Figures and Tables

**Table 1 nutrients-17-02706-t001:** Summary of the data for each SS according to the organizations and research cited above.

Supplement	General Dosage	Sports	Protocol and Dosage	Main Effect	Commercial Format
Caffeine	Minimum 3 mg per kg BW 3–6 mg per kg BW	Endurance sports. Intermittent and team sports Swimming and rowing Strength–power sports	Administer pure caffeine/anhydrous 60 min before exercise	Improves sports performance in aerobic and anaerobic exercises Reduces the perception of effort during exercise Increases endurance capacity	Capsule, beverages, gels, shots, chewing gum, powders
Creatine	2 ways: 1. Loading phase + maintenance phase 2.Maintenance phase Loading phase: 20–30 g per day or 0.3 g per kg BW for 5–7 days and distributed in 4 intakes during the main meals Maintenance phase: 3–5 g per day or 0.03 g per kg BW per day	Endurance sports Intermittent and team sports Swimming and rowing Strength–power sports	Administer as creatine monohydrate Before or after training Accompany with 50 g of protein and carbohydrates	Improves sports performance Increases physical performance in successive intervals of short-term, high-intensity exercise Daily creatine intake can improve the effect of resistance training on muscle strength in adults over 55 years of age	Capsules, tablets, powder
Beta alanine	4–6 g per day, distributed in different doses per day	Short duration sports Short-term sports Intermittent and team sports	Take for 4 to 10 weeks	Improved sports performance	Tablets, powder, capsules, shots
Nitrates/beetroot juice	300–600 mg or 5–8 mmol	Endurance sports Intermittent and team sports Sports whose duration is less than 40 min	Take 2–3 h before exercise	Improved sports performance	Powder, gel, shot, liquid, tablets, capsules
Sodium bicarbonate	0.2–0.4 g per Kg BW	Endurance sports Intermittent and team sports Swimming Sport with weight category Short duration sports	Two routes of administration: Take 60 to 120 min before exercise as a single dose Take the dose in 3–4 intakes per day, during the 2–4 days prior to the competition	Improved sports performance	Powder, tablets
Glycerol	1–1.2 g per kg BW together with 25 mL per kg BW of liquid	Endurance sports Long-term sports Sports in hot conditions	Take between 1 and 2 h before exercise	Improved sports performance Helps combat dehydration	Powder

BW: body weight, g: grams, mg: milligrams, mmol: millimoles, kg: kilograms, h: hour.

## Data Availability

The data presented in this study are available in the tables of this article.

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
