# Peer review of "Ergogenic and Physiological Effects of Sports Supplements: Implications for Advertising and Consumer Information"

_nutrients, 2025, doi:10.3390/nu17162706_

Round 1
Reviewer 1 Report
Comments and Suggestions for Authors
This study evaluates the ergogenic and physiological effects of Category A sports supplements (e.g., caffeine, creatine, β-alanine) and assesses their regulatory compliance and advertising practices in the EU, highlighting gaps between evidence and approved health claims. The authors concluded that while caffeine and creatine have EFSA-approved claims, supplements like β-alanine and nitrates lack regulatory approval despite strong evidence. The paper calls for updated labeling standards, ethical marketing, and harmonized EU regulations to protect consumers and support evidence-based use in sports nutrition.
The manuscript is well written and present interesting conclusions. However, some modifications are needed:
I suggest that the following papers are helpful in the context of the present study:
Salem et al., (2025). Acute effects of beetroot juice vs. creatine supplementation on maximal strength, autonomic regulation, and muscle oxygenation during incremental resistance exercise. Biology of Sport, 42(4), 241-259.
Bougrine et al., N. (2024). Effects of Different Caffeine Dosages on Maximal Physical Performance and Potential Side Effects in Low-Consumer Female Athletes: Morning vs. Evening Administration. Nutrients, 16(14), 2223.
Title. The current title is lengthy and could be more concise. Example:
"Ergogenic and Physiological Effects of Sports Supplements: Implications for Advertising and Consumer Information"
Keywords. Add "regulatory framework" and "athletic performance" to improve searchability.
Introduction. The novelty of the paper needs clearer presentation. Explain what gap in literature this paper fills.
Methods. Indicate how many papers were included in the final synthesis.
Methods. Clarify the PubMed search terms (e.g., "sports supplements AND meta-analysis")
Methods. Specify data extraction methods more clearly — was it conducted independently by reviewers? Any bias mitigation strategy?
Limitations. Acknowledge potential publication bias in meta-analyses and the exclusion of non-English studies.
Tables. Use footnotes and legends for abbreviations (e.g., BW, CH, SS).
Remove typographical errors (e.g., extra punctuation, awkward spacing, line breaks)
Other minor changes:
- “maximing” → “maximizing”
- “conotations in values” → “connotations of values”
- “Fitness Influencers” → should not be capitalized unless part of a formal term.
- “sodium carbonate” → should be “sodium bicarbonate” unless referring to something else intentionally.
- Define acronyms (e.g., AIS, ISSN) at first use.
- check "Aguilar & Arbaiza, 2021" formatting
Author Response
REVIEWER 1
This study evaluates the ergogenic and physiological effects of Category A sports supplements (e.g., caffeine, creatine, β-alanine) and assesses their regulatory compliance and advertising practices in the EU, highlighting gaps between evidence and approved health claims. The authors concluded that while caffeine and creatine have EFSA-approved claims, supplements like β-alanine and nitrates lack regulatory approval despite strong evidence. The paper calls for updated labeling standards, ethical marketing, and harmonized EU regulations to protect consumers and support evidence-based use in sports nutrition.
The manuscript is well written and present interesting conclusions. However, some modifications are needed:
I suggest that the following papers are helpful in the context of the present study:
- Salem et al., (2025). Acute effects of beetroot juice vs. creatine supplementation on maximal strength, autonomic regulation, and muscle oxygenation during incremental resistance exercise. Biology of Sport, 42(4), 241-259.
- Bougrine et al., N. (2024). Effects of Different Caffeine Dosages on Maximal Physical Performance and Potential Side Effects in Low-Consumer Female Athletes: Morning vs. Evening Administration. Nutrients, 16(14), 2223.
Response of the authors: We appreciate the reviewers' comments. The references have been added in the discussion, in the paragraphs 2 and 10 respectively.
Title. The current title is lengthy and could be more concise. Example:
"Ergogenic and Physiological Effects of Sports Supplements: Implications for Advertising and Consumer Information"
Response of the authors: We appreciate the reviewers' comments and we have included the tittle as the reviewer recommends.
Keywords. Add "regulatory framework" and "athletic performance" to improve searchability.
Response of the authors: We appreciate the reviewers' comments and the keywords has been included.
Introduction. The novelty of the paper needs clearer presentation. Explain what gap in literature this paper fills.
Response of the authors: We appreciate the reviewers' comments. To address this, we have revised the end of the first paragraph of the Introduction to more clearly articulate the novelty of our study. Specifically, we now explain that although health and nutrition claims in the EU require prior approval by the European Commission based on EFSA’s scientific opinion, many companies use unapproved claims—particularly in food supplements—during the interim. This regulatory gap is underexplored in the literature. Our study aims to fill this gap by analyzing the alignment between these commercial claims and scientific evidence
Methods. Indicate how many papers were included in the final synthesis.
Response of the authors: We appreciate the reviewers' comments, The wording of the methodology has been improved to clarify the selection of studies. Although not a formal systematic review, our methodology ensured the inclusion of high-quality and up-to-date evidence.
Methods. Clarify the PubMed search terms (e.g., "sports supplements AND meta-analysis")
Response of the authors: This information has been described in subsection 2.2. Sample selection strategy “In addition, recent systematic reviews with meta-analysis were included through the PubMed database, using as keyword the name of the supplement (e.g., “caffeine”) and applying the systematic review and meta-analysis filter.”
Methods. Specify data extraction methods more clearly — was it conducted independently by reviewers? Any bias mitigation strategy?
Response of the authors: We appreciate the reviewers' comments, subsection 2.4 has been expanded to clarify this aspect.
Limitations. Acknowledge potential publication bias in meta-analyses and the exclusion of non-English studies.
Response of the authors: We appreciate the reviewers' comments. The limitation has been added in the “Limitations of the study”. Also, The limitations subsection has been improved to clarify the aspects mentioned by the reviewer.
Tables. Use footnotes and legends for abbreviations (e.g., BW, CH, SS).
Remove typographical errors (e.g., extra punctuation, awkward spacing, line breaks)
Response of the authors: We appreciate the reviewers' comments and we have added some abbreviations that left in the Table 7
Other minor changes:
- “maximing” → “maximizing”
- “conotations in values” → “connotations of values”
- “Fitness Influencers” → should not be capitalized unless part of a formal term.
- “sodium carbonate” → should be “sodium bicarbonate” unless referring to something else intentionally.
- Define acronyms (e.g., AIS, ISSN) at first use.
- check "Aguilar & Arbaiza, 2021" formatting
Response of the authors: We appreciate the reviewers' comments. All the changes has been done, except fitness influencers, which is part of a formal term that is use in other papers that we used and cited.
About the acronyms all are defined at the first use (normally in the abstract or introduction) and after in “Abbreviations” part at the end of the document.
Reviewer 2 Report
Comments and Suggestions for Authors
This is a timely and well-structured review that explores the ergogenic and physiological effects of sports supplements classified as Category A by the Australian Institute of Sport. The topic is both relevant and important, particularly given the widespread use of supplements and the confusion that often surrounds health claims and marketing.
The paper is based on solid sources, including consensus statements and systematic reviews, and provides a comprehensive summary of dosage, protocols, effects, and regulatory aspects. The tables are detailed, clear, and helpful for both researchers and practitioners.
Recommedations
There are some grammatical and phrasing issues that make the text harder to follow in places. For example:
In the introduction, the phrase “by the hand of those known as ‘Fitness Influencers’”(line 85, page 3) could be reworded more naturally to “through the use of fitness influencers.” Also, some long or overly formal sentences could be shortened or simplified.
While the manuscript summarizes the findings of consensus statements and systematic reviews, it would benefit from a brief discussion of the limitations of the available evidence. E.g. in the paper, it mentions that many systematic reviews include studies with small sample sizes or high variability in protocols. Also, some effects (e.g., of β-alanine or glycerol) are modest and context-dependent, yet often presented as generalizable.
Even a short paragraph reflecting on these issues would strengthen the interpretation of results.
The authors briefly mention variability in supplement responses, but this deserves more attention. For example, caffeine metabolism differs significantly between individuals depending on genetics (e.g., CYP1A2 polymorphism), which can affect its ergogenic effect or even lead to adverse reactions. Moreover, female athletes may respond differently to creatine or β-alanine supplementation depending on hormonal status, yet most studies include mostly male participants. Expanding this section would help nuance the recommendations.
Comments on the Quality of English Language
In the introduction, the phrase “by the hand of those known as ‘Fitness Influencers’”(line 85, page 3) could be reworded more naturally to “through the use of fitness influencers.” Also, some long or overly formal sentences could be shortened or simplified.
Author Response
REVIEWER 2
This is a timely and well-structured review that explores the ergogenic and physiological effects of sports supplements classified as Category A by the Australian Institute of Sport. The topic is both relevant and important, particularly given the widespread use of supplements and the confusion that often surrounds health claims and marketing. The paper is based on solid sources, including consensus statements and systematic reviews, and provides a comprehensive summary of dosage, protocols, effects, and regulatory aspects. The tables are detailed, clear, and helpful for both researchers and practitioners.
Recommedations
There are some grammatical and phrasing issues that make the text harder to follow in places. For example:
In the introduction, the phrase “by the hand of those known as ‘Fitness Influencers’”(line 85, page 3) could be reworded more naturally to “through the use of fitness influencers.” Also, some long or overly formal sentences could be shortened or simplified.
Response of the authors: We appreciate the reviewers' comments. That example has been changed following the reviewer vision. Also, the text has been re-valued to do it shorter and simplifier
While the manuscript summarizes the findings of consensus statements and systematic reviews, it would benefit from a brief discussion of the limitations of the available evidence. E.g. in the paper, it mentions that many systematic reviews include studies with small sample sizes or high variability in protocols. Also, some effects (e.g., of β-alanine or glycerol) are modest and context-dependent, yet often presented as generalizable.
Response of the authors: We appreciate the reviewers' comments. We added that limitation in the “Limitations of the study”
Even a short paragraph reflecting on these issues would strengthen the interpretation of results.
Response of the authors: We appreciate the reviewers' comments. We added to the last part of the practical applications this point
The authors briefly mention variability in supplement responses, but this deserves more attention. For example, caffeine metabolism differs significantly between individuals depending on genetics (e.g., CYP1A2 polymorphism), which can affect its ergogenic effect or even lead to adverse reactions. Moreover, female athletes may respond differently to creatine or β-alanine supplementation depending on hormonal status, yet most studies include mostly male participants. Expanding this section would help nuance the recommendations.
Response of the authors: We appreciate the reviewers' comments, we have extended this part in the discussion in the second and fifth paragraph, we also added the references to support them.
Reviewer 3 Report
Comments and Suggestions for Authors
Submitted for review is a manuscript on Claims Regarding The Ergogenic and Physiological Effects of Sports Supplements, and Their Possible Implications for Advertising and Consumer Information. already in the title contains the problem of marketing ergogenic agents and dietary supplements. The study presented is an expert opinion on the structure, marketing features and manufacturers' recommendations for the use of AIS category A supplements. The authors write “...the aim of this study is to describe and compile the ergogenic and physiological effects of the most evidenced sports supplements in relation to position papers and systematic reviews.” The question must then be raised as to where the scientific problem, the actual research objective of the manuscript, is located. If we then read that “With all the information gathered, a series of guidelines will be proposed, by way of suggestions-guidelines, of the implications and potential applications that this work may have for the elaboration of messages in the context of advertising communication and consumer protection.”, then I know that we are dealing with a project of very low scientific significance. The work what the authors emphasize is to bring information“ in the context of advertising communication and consumer protection”. The importance and need for such work cannot be denied, of course. However, the context, according to the reviewer, is far from the profile of the journal and from the criteria of scientific papers. After reading the entire manuscript, one becomes convinced that the most scientific “element” of this study is the design of the analysis of source materials. The knowledge that we take away after reading the paper is devoid of new information that contributes to the knowledge of dietary supplements for athletes. Conclusions in the manuscript boil down to indicating for whom the effects of marketing analyses can be useful, and points to common knowledge like: " although there are some variations in administration protocols and observed effects, mainly attributable to differences between populations, methodologies and sports disciplines analyzed . These variations are especially notable in supplements such as β-alanine and sodium bicarbonate...". The statement “It can also serve as a use ful resource for dietitians-nutritionists and coaches who require a general guide based on the supplements ...” indicates that we are not dealing with a scientific study, a research paper, but rather a guide for dietitians and coaches. In the context of the comments presented above, I do not recommend the manuscript for publication in a scientific journal due to the lack of content that significantly enriches the knowledge in the field of supplementation for athletes. The work should be directed to publication in journals addressing marketing and formal-legal issues related to the distribution of supplements for athletes.
Author Response
REVIEWER 3
Submitted for review is a manuscript on Claims Regarding The Ergogenic and Physiological Effects of Sports Supplements, and Their Possible Implications for Advertising and Consumer Information. already in the title contains the problem of marketing ergogenic agents and dietary supplements. The study presented is an expert opinion on the structure, marketing features and manufacturers' recommendations for the use of AIS category A supplements. The authors write “...the aim of this study is to describe and compile the ergogenic and physiological effects of the most evidenced sports supplements in relation to position papers and systematic reviews.” The question must then be raised as to where the scientific problem, the actual research objective of the manuscript, is located.
If we then read that “With all the information gathered, a series of guidelines will be proposed, by way of suggestions-guidelines, of the implications and potential applications that this work may have for the elaboration of messages in the context of advertising communication and consumer protection.”, then I know that we are dealing with a project of very low scientific significance. The work what the authors emphasize is to bring information“ in the context of advertising communication and consumer protection”. The importance and need for such work cannot be denied, of course. However, the context, according to the reviewer, is far from the profile of the journal and from the criteria of scientific papers. After reading the entire manuscript, one becomes convinced that the most scientific “element” of this study is the design of the analysis of source materials. The knowledge that we take away after reading the paper is devoid of new information that contributes to the knowledge of dietary supplements for athletes. Conclusions in the manuscript boil down to indicating for whom the effects of marketing analyses can be useful, and points to common knowledge like: " although there are some variations in administration protocols and observed effects, mainly attributable to differences between populations, methodologies and sports disciplines analyzed . These variations are especially notable in supplements such as β-alanine and sodium bicarbonate...".
The statement “It can also serve as a use ful resource for dietitians-nutritionists and coaches who require a general guide based on the supplements ...” indicates that we are not dealing with a scientific study, a research paper, but rather a guide for dietitians and coaches. In the context of the comments presented above, I do not recommend the manuscript for publication in a scientific journal due to the lack of content that significantly enriches the knowledge in the field of supplementation for athletes. The work should be directed to publication in journals addressing marketing and formal-legal issues related to the distribution of supplements for athletes.
Response of the authors: We appreciate the reviewers' comments for the time and effort dedicated to evaluating our manuscript and for providing a detailed and critical assessment. We respectfully acknowledge the concerns raised and would like to offer several clarifications in defense of the scientific and practical value of our work.
First, although the manuscript does not follow the structure of an original experimental study, it is structured as a narrative review with practical applications, based on a systematic synthesis of high-level evidence from position statements and recent systematic reviews with meta-analyses. This methodological framework has been explicitly clarified and justified in the revised version of the manuscript, as suggested by Reviewers. We have also expanded the “Materials and Methods” and “Limitations” sections to reflect this more transparently.
Second, we agree that our approach is applied and oriented to real-world impact, especially in the areas of advertising communication and consumer protection. However, we would argue that this does not diminish its scientific value, but rather aligns with growing interest in translational research, particularly in journals such as Nutrients, where the intersection between science, regulation, industry practices, and public health is increasingly relevant. Indeed, the work addresses an identified gap: although the AIS Category A supplements are well studied in isolation, there is no integrated, evidence-based resource that links their ergogenic effects, dosage protocols, and regulatory considerations in a way that can inform both clinical professionals and the supplement industry. Our aim was not to re-demonstrate the efficacy of these supplements, already well established in the literature, but to provide a structured, practical synthesis that facilitates the accurate use of health claims, product reformulation, and scientifically sound marketing strategies, especially in the European context.
We also highlight that the authors of this manuscript have previously published related work in Nutrients addressing regulatory, labeling, and advertising issues in sports supplements, contributing to a line of research recognized by the journal. For example:
- https://pubmed.ncbi.nlm.nih.gov/29117104/
- https://pubmed.ncbi.nlm.nih.gov/33917477/
- https://pubmed.ncbi.nlm.nih.gov/33468268/
- https://pubmed.ncbi.nlm.nih.gov/40507192/
- https://pubmed.ncbi.nlm.nih.gov/38999728/
These publications reflect the novelty and continuity of a scientific line that connects nutritional evidence with legal and communication aspects of public health relevance.
Moreover, Table 7 in the current manuscript represents a valuable innovation, summarizing dosing, protocols, effects, and formats in a single integrated source. This table is not a mere collation of existing data but a structured harmonization that can directly support:
- Reformulation of supplement products.
- Evidence-based marketing claims.
- Consumer transparency and labeling strategies.
- Finally, we note that we have addressed and incorporated substantial improvements suggested by the other three reviewers, including:
- Clarifying the manuscript type and objective.
- Improving the structure and readability of the tables.
- Adding new content to the Discussion and Limitations sections, including critical perspectives on modest effects, inter-individual variability, and potential commercial bias.
We hope that this clarification helps reframe the manuscript not as a “low scientific significance” project, but rather as a targeted and rigorously developed resource that bridges science and application, consistent with the broader mission of Nutrients to support professionals and stakeholders in the field of nutrition and health.
Reviewer 4 Report
Comments and Suggestions for Authors
Thank you for the opportunity to review the manuscript entitled “Claims Regarding The Ergogenic and Physiological Effects of Sports Supplements, and Their Possible Implications for Advertising and Consumer Information”.
This manuscript offers a comprehensive and practical review of Category A sports supplements (as classified by the AIS), focusing on their ergogenic and physiological effects, and critically addressing the advertising and regulatory implications, especially within the European framework. The work is timely, informative, and bridges an important gap between scientific evidence, regulatory constraints, and consumer communication.
Suggestions for authors:
- Manuscript Type. The authors classify this submission as an article, yet the structure and content show it is more accurately a narrative review with practical applications. It would be helpful for the authors to clearly define whether this is intended as a narrative review, a scoping review, or another specific type, as this distinction affects how the content is interpreted and evaluated.
- Table Clarity. The tables provided are overly dense and visually difficult to interpret. I recommend improving their readability by:
- Missing Key Information in Tables. To improve the scientific value and transparency of the manuscript, I suggest the authors:
- Include the number of athletes involved in the cited studies (when available),
- Please add a short comment about inter-individual variability (e.g., gender, age, training status) in supplement efficacy and tolerance. This is particularly relevant for β-alanine and caffeine. Indicate sex-specific responses, as physiological differences between male and female athletes may significantly influence outcomes.
- Balance in Reporting Results. To provide a more objective and comprehensive synthesis, the manuscript should also include studies that found no significant performance effects from supplementation. Presenting only positive outcomes may introduce confirmation bias and limit the critical scope of the review.
- Scope and Limitations. While focusing on AIS Category A supplements is understandable, please briefly justify the exclusion of other categories and acknowledge that this limits the review’s generalizability to emerging supplements. The section on limitations could also mention potential commercial bias in studies or marketing materials analyzed, and the impact of underreporting of adverse effects. Consider including emerging supplements (even in an appendix).
- Abstract. Conclusions. It's more correct to end this way….“while performance supplements may enhance athletic performance when used alongside proper nutrition and scientific guidance, their effectiveness is not always consistent or assured”.
Author Response
REVIEWER 4
Thank you for the opportunity to review the manuscript entitled “Claims Regarding The Ergogenic and Physiological Effects of Sports Supplements, and Their Possible Implications for Advertising and Consumer Information”. This manuscript offers a comprehensive and practical review of Category A sports supplements (as classified by the AIS), focusing on their ergogenic and physiological effects, and critically addressing the advertising and regulatory implications, especially within the European framework. The work is timely, informative, and bridges an important gap between scientific evidence, regulatory constraints, and consumer communication.
Suggestions for authors:
Manuscript Type. The authors classify this submission as an article, yet the structure and content show it is more accurately a narrative review with practical applications. It would be helpful for the authors to clearly define whether this is intended as a narrative review, a scoping review, or another specific type, as this distinction affects how the content is interpreted and evaluated.
Response of the authors: We thank the reviewer for this valuable observation. We agree that the manuscript, as currently structured and presented, more accurately fits the characteristics of a narrative review with practical applications, rather than an original research article. Accordingly, we have updated the classification of the manuscript type to “Narrative Review” to better reflect the nature of the work.
We have also clarified this in the revised version of the manuscript, specifically in the first paragraph of the Materials and Methods section, where we now describe the article as a narrative review built upon predefined criteria for supplement selection and document inclusion, based on authoritative sources such as systematic reviews and position statements. This clarification will help readers and evaluators better understand the methodological framework and scope of the manuscript.
Table Clarity. The tables provided are overly dense and visually difficult to interpret. I recommend improving their readability by:
Missing Key Information in Tables. To improve the scientific value and transparency of the manuscript, I suggest the authors:
Include the number of athletes involved in the cited studies (when available),
Response of the authors: We appreciate the reviewers' comments. Regarding the readability, we acknowledge that the tables include a substantial amount of information. However, due to the applied and comparative nature of the manuscript, we considered it important to present in a single view the protocols of use, dosages, effects, sports involved, and sources for each supplement. That said, we have revised the format of the tables to improve their visual clarity, including adjustments such as spacing, alignment, and the use of subheadings where appropriate to facilitate interpretation.
As for the suggestion to include the number of athletes involved in the cited studies, we respectfully note that this is not feasible within the scope of our work. The manuscript is a narrative review that synthesizes information from systematic reviews and position statements, rather than from individual primary studies. Consequently, the number of participants from original research was not extracted or considered during our synthesis. Our objective was to provide a practical summary of the most established protocols and effects, based on aggregated high-level evidence, rather than conduct a detailed analysis of individual study characteristics.
Please add a short comment about inter-individual variability (e.g., gender, age, training status) in supplement efficacy and tolerance. This is particularly relevant for β-alanine and caffeine. Indicate sex-specific responses, as physiological differences between male and female athletes may significantly influence outcomes.
Response of the authors: We appreciate the reviewers' comments. We have extended this part in the discussion in the second and fifth paragraph, we also added the references to support them. For Caffeine, Beta alanine and also for Creatine, because we considere the most important supplement influenced by genetic, sex… and because other reviewer did the same comment
Balance in Reporting Results. To provide a more objective and comprehensive synthesis, the manuscript should also include studies that found no significant performance effects from supplementation. Presenting only positive outcomes may introduce confirmation bias and limit the critical scope of the review.
Response of the authors: We appreciate the reviewers' comments We have used recent reviews and the results are generally positive because we are relying on supplements classified as GROUP A by the Australian Institute of Sport, so there is a good deal of evidence to support their use in the contexts specified. It is therefore normal and understandable that recent reviews have mainly favourable results. Nonetheless, we have revised the Discussion section to acknowledge that not all studies within the included reviews found significant or consistent effects, and that outcomes may vary depending on individual response, training status, or sport discipline. We have also emphasized that some supplements—such as β-alanine and glycerol—show modest or context-dependent effects, and that results are not universally generalizable.
Scope and Limitations. While focusing on AIS Category A supplements is understandable, please briefly justify the exclusion of other categories and acknowledge that this limits the review’s generalizability to emerging supplements. The section on limitations could also mention potential commercial bias in studies or marketing materials analyzed, and the impact of underreporting of adverse effects. Consider including emerging supplements (even in an appendix).
Response of the authors: We appreciate the reviewers' comments. We have added a more explicit justification for the exclusion of AIS Category B (emerging) supplements in the “Sample selection strategy” section, and we have reflected this decision as a limitation in the “Limitations of the study” section. In addition, we have incorporated references to potential commercial bias in the included studies and the underreporting of adverse effects, as the reviewer recommended.
We would also like to clarify that the decision to focus solely on AIS Category A performance supplements was deliberate and aligned with the objective of the manuscript: to summarize and systematize the protocols, dosages, effects, and regulatory aspects of supplements with strong scientific evidence, used under validated protocols and recognized by international institutions for their effectiveness and safety. These supplements are considered appropriate for use by identified athletes in specific performance contexts and are backed by well-established recommendations.
In contrast, Category B supplements represent emerging evidence and are only recommended for use within research protocols or under individual monitoring, which would fall outside the scope and aims of this manuscript. While we recognize the growing interest in these emerging substances, we believe that their inclusion would dilute the practical focus of the review and introduce considerable uncertainty regarding protocols, efficacy, and regulatory status.
It is also worth noting that some of our prior research has demonstrated that the advertising claims used by companies—even for Category A supplements—are often exaggerated and not aligned with scientific evidence or with health claims approved by the European Commission. This issue becomes even more problematic with Category B supplements, given the lack of regulatory consensus and weaker scientific backing.
We have opted not to include an appendix for emerging supplements to maintain a clear and focused structure centered on validated supplements with high evidence, but we appreciate the reviewer’s suggestion and may consider it for future work specifically addressing novel or unregulated products.
Some of the authors' works that cover this topic can be found in:
- https://pubmed.ncbi.nlm.nih.gov/33917477/
- https://pubmed.ncbi.nlm.nih.gov/33468268/
- https://pubmed.ncbi.nlm.nih.gov/33468268/
Abstract. Conclusions. It's more correct to end this way….“while performance supplements may enhance athletic performance when used alongside proper nutrition and scientific guidance, their effectiveness is not always consistent or assured”.
Response of the authors: We appreciate the reviewers' comments. We finished the conclusions in abstract in that way.
Round 2
Reviewer 3 Report
Comments and Suggestions for Authors
Since the authors consider marketing-oriented works to be a scientific contribution to the development of knowledge about supplementation, we cannot find common ground. However, I have a question in response to the authors' comments on my remarks: what is the scientific contribution of Table 7? I would like to point out that I do not evaluate works based on other publications by the authors in the same journal. In my opinion, such reasoning is unacceptable and is not part of scientific discussion. I understand that the Special Issue Effects of Exercise and Diet on Health is to be filled with the work key words education. But there is no education here because the overly broad restrictions do not allow this task to be fulfilled.
Author Response
Response of the authors: We appreciate the reviewers' comments for the time and effort dedicated to evaluating our manuscript and for providing a detailed and critical assessment.
Regarding your question about the scientific contribution of Table 7, we would like to clarify its purpose and methodological basis. Table 7 presents a synthesis of key findings derived from systematic reviews and position statements issued by leading international organizations (e.g., IOC, ISSN, AIS, EFSA). This summary was developed through a structured process that included expert consensus, and it aims to consolidate practical, evidence-based recommendations on the use of performance supplements. Rather than replicating existing literature, this table distills fragmented information into an accessible, harmonized format that can be used by health professionals, athletes, and stakeholders in the sports nutrition and communication fields.
Furthermore, given that certain supplements (e.g., β-alanine, sodium bicarbonate, nitrates, glycerol) there is no scientific opinion from EFSA, and no health claims have been approved by the European Commission despite having strong scientific support, Table 7 offers a valuable framework for proposing responsible claims and conditions of use based on the current evidence. In this way, the table contributes to bridging the gap between scientific knowledge and regulatory or commercial application.
With respect to your concern about the educational value of the manuscript, we understand and respect your position. Our intention was to provide an informative resource that supports professionals in making informed decisions about supplement communication, contributing to consumer protection and evidence-based marketing. While this may not constitute formal educational content in a traditional sense, we believe it has an educational role in promoting critical understanding and responsible application of scientific data.
Lastly, we acknowledge your comments regarding the reference to other publications by the authors. Our intention was not to justify the merit of this work based on previous publications, but rather to contextualize the line of research within which this manuscript is framed—specifically, one that explores the interface between evidence-based supplementation, regulatory frameworks, and advertising communication.
Once again, we appreciate your review and the opportunity to clarify these points.
Reviewer 4 Report
Comments and Suggestions for Authors
The authors responded to the questions and comments raised, and their responses are quite satisfactory.The authors are still labeling the manuscript as an ARTICLE in the journal template, which should be corrected.
The tables remain somewhat difficult to read. I believe the manuscript would benefit from these revisions before it can be considered for publication.
Author Response
Response of the authors: We thank the reviewer´ comments. As per your suggestion, to improve readability and reduce visual overload in the main text, we have moved the detailed tables (previously Tables 1–6) to the end of the manuscript as Appendices (Tables A1–A6). Only the synthesized summary table, now labeled Table 1, remains in the main body of the manuscript. We believe this improves the clarity and flow of the review while maintaining full access to the detailed data for interested readers.